# Time to major adverse drug reactions and its predictors among children on antiretroviral treatment at northwest Amhara selected public hospitals northwest; Ethiopia, 2023

**Bantegizie Senay Tsega[1], Abebe Habtamu[2], Moges Wubie[2], Animut Takele Telayneh[2], Bekalu Endalew[2], Samuel Derbie Habtegiorgis[2], Molla Yigzaw Birhanu[2], Worku Misganaw Kebede[3], Keralem Anteneh Bishaw[4]***

1 Fintoselam Hospital, Obstetrics & Gynecology Ward, Fintoselam, Ethiopia, 2 Department of Public Health, College of Medicine and Health Sciences, Debre Markos University, Debre Markos, Ethiopia, 3 Department of Nursing, College of Medicine and Health Sciences, Debre Markos University, Debre Markos, Ethiopia, 4 Department of Midwifery, College of Medicine and Health Sciences, Debre Markos University, Debre Markos, Ethiopia

* keralemante2010@gmail.com, keralem_anteneh@dmu.edu.et

**Data Availability Statement:** All relevant data are within the paper and its Supporting Information.

## Abstract

### Background

Adverse drug reaction is one of the emerging challenges in antiretroviral treatment. Determining the incidence rate and predictors among children on antiretroviral treatment (ART) is essential to improve treatment outcomes and minimize harm. And also, evidence regarding the time to major adverse drug reactions and its predictors among children on antiretroviral treatment is limited in Ethiopia.

### Objective

This study aimed to assess the time to major adverse drug reaction and its predictors among children on antiretroviral treatment at selected public hospitals in Northwest Amhara, Ethiopia, 2023.

### Method

A retrospective cohort study was conducted among 380 children on antiretroviral treatment who enrolled from June 27, 2017, to May 31, 2022. Data was collected using a structured data extraction checklist. Data were entered into Epidata 4.6 and analyzed using STATA 14. The incidence rate of major adverse drug reactions was determined per person/months. The Cox proportional hazards regression model was used to identify predictors of major adverse drug responses. A p-value less than 0.05 with a 95% CI was used to declare statistical significance.

**Funding:** The author(s) received no specific funding for this work.

**Competing interests:** The author(s) received no specific funding for this work.

## Result

The minimum and maximum follow-up time was 6 and 59 months, respectively. The study participants were followed for a total of 9916 person-months. The incidence rate of major adverse drug reactions was 3.5 /1000 person–months. Advanced clinical stages of HIV/AIDS (III and IV) [adjusted hazard ratio = 7.3, 95% CI: 2.74–19.60)], poor treatment adherence [adjusted hazard ratio = 0.33, 95% CI: 0.21–0.42], taking antiretroviral treatment twice and more [adjusted hazard ratio = 3.43, 955 CI: (1.26–9.33)] and not taking opportunistic infection prophylaxis [adjusted hazard ratio = 0.35, 95% CI: 0.23–0.52)] were predictors of major adverse drug reactions.

## Conclusion

The incidence rate of major adverse drug reactions among children on antiretroviral treatment was congruent with studies in Ethiopia. Advanced clinical stages of HIV/AIDS, poor treatment adherence, taking antiretroviral treatment medications twice or more, and not taking opportunistic infection prophylaxis were predictors of major adverse drug reactions.

## Introduction

Globally, more than 2 million children worldwide are infected with HIV; approximately 90% of them live in Sub-Saharan Africa. Sub-Saharan Africa has the highest burden of HIV/AIDS in the world. It remained the most severely affected region, accounting for 71% of all new HIV infections, with an estimated 430,000 new HIV infections among children less than 15 years [1–3].

In Ethiopia, the national HIV prevalence was 0.9, and 21,146 children < 15 years are taking antiretroviral therapy [4]. Infants and young children infected with HIV have exceptionally higher morbidity and mortality. Without intervention, up to 52% and 75% of children die before the age of two (2) and five (5) years, respectively. And also, when children begin ART therapy, severe medication reactions and poor adherence are significant challenges [4].

Adverse drug reactions (ADRs) are undesirable effects of drugs administered at recommended doses via the recommended method of administration for prevention and treatment [5]. It affected both the healthcare system and the patients. ART-related side effects range from acute to potentially fatal adverse effects of ART and often need an emergency cessation of drugs [6]. ADRs of antiretroviral (ARV) drugs and other medicines are recognized as the primary cause of mortality among people with HIV/AIDs [7]. ADRs are common in ART patients, leading to treatment failure, regimen modifications, poor adherence, and treatment discontinuation [8]. Up to 25% of patients discontinue their initial ART regimen during the first eight months of therapy due to ADRs [9].

Adverse drug reactions (ADRs) can affect the efficacy of antiretroviral therapy in children in several ways. It can result in treatment interruptions or discontinuance and reduce treatment adherence, which leads to viral rebound and drug resistance. ADRs can damage organs or cause other major health issues, which require additional medical interventions and increase healthcare costs. It affects children's and families quality of life, including physical discomfort, emotional stress, and social isolation [10, 11].

A study in Namibia found that the earliest adverse effect was Neverapine-induced rash, with onset times averaging seven (7) days and ranging from five (5) days to two (2) weeks after

the ART drug started. Hepatotoxicity was initially observed on average 1.7 years after starting antiretroviral treatment, with a range of 1 month to 5 years. Zidovudine (AZT) caused anemia, with a mean hemoglobin drop of 4.2–2.3 gm/dl [12]. A study in Nigeria also found that anemia and skin rash were the most typical ADRs observed. Almost 45% of ADRs occurred within the first three months of treatment [13]. Furthermore, clinical hepatitis, peripheral neuropathy, and lipodystrophy are reported as ADRs of ART [14].

The incidence rate of ADRs among HIV patients ART was 4.1 per 100 person-years [15]. A study in Ethiopia reported that more than ninety percent (90.74%) of participants developed ADRs within one (1) year. It was most frequently reported in patients with advanced WHO clinical stage of HIV, lower body mass index (BMI), those with poor adherence, and Tenofovir (TDF)—Lamivudine (3TC)-Efavirenz (EFV) regimen [16].

Many factors affect the incidence of adverse drug reactions, but the most significant ones were a decrease lower CD4 cell count, advanced WHO HIV/AIDS clinical stages, not taking opportunistic infection prophylaxis, a Tenofovir (TDF)–Nevirapine (NVP) containing regimen, being bedridden at the time of treatment initiation, and concurrent treatment administration [17–20].

WHO revised ART guidelines multiple times, and ADRs was the main reason [21, 22]. Evidence regarding time to major adverse drug reactions (MADRs) and its predictors in Ethiopia for Dolutegravir (DTG)-based ART regimens is scarce. And also, there is no published study in the study setting. Therefore, the study aimed to assess the time to MADRs incidence and its predictors among HIV-positive children on antiretroviral therapy at selected public hospitals in Northwest Ethiopia.

## Methods

### Study area, setting, and population

This hospital-based retrospective cohort study was conducted at selected public hospitals of the northwest Amhara region from June 27, 2017, to May 31, 2022, with ART services beginning after the 2017 G.C. The study included Felege Hiwot Comprehensive Specialized Hospital (FHCSH), Adet Primary Hospital, Finote Selam General Hospital, and Debre Markos Comprehensive Specialized Hospital (DMCSH). The study settings are 565km, 510km, 385km, and 299 km, respectively, away from Addis Ababa, the capital city of Ethiopia. The 2022 zonal report and the ART registry logbook reported that FHCSH, Adet Primary Hospital, Finote Selam General Hospital, and DMCSH provided ART for 6984, 221, 2052, and 3880 patients, respectively. In all selected hospitals, trained General practitioners, nurses, and public health officers provide ART services. The study's source population included all children under the age of 15 years who were receiving ART services at public hospitals in Northwest Amhara. All under 15 children who had ART follow-up at selected northwest Amhara region public hospitals from June 27, 2017, to May 31, 2022, were the study population. Patients with incomplete medical records, such as regimen type and date of enrolment on ART, were excluded from the study. The data extraction period was from February 15 to March 30/2023.

### Sampling size determination

The sample size was determined using the survival analysis formula by considering the assumption of 95% CI and a study power of 80%, Hazard rate (3.2) from the previous study at Bahirdar, Ethiopia [23]. The number of events (E) calculated using the formula = (Za/2+zB) 2/ (PQ) (logHR) 2, where = Za/2 = 1.96, ZB = 0.84, HR = 3.2, p = proportion of subjects for the exposed group is = 0.5 in the exposed group, and Q = proportion of subjects in unexposed group = 0.5 according to freedman principle. Based on the calculation, the event was 24. The

sample size was determined using the formula = Event (E) / P (E), where P (E) = probability of an event (0.098) from an input of the freedman principle. The study's final sample size was 402 after considering the replacement of missed medical records and the design effect (1.5).

## Sampling technique and sampling procedure

A multi-stage sampling procedure was used in the study. First, a simple random sampling (lottery) method was used to select from all public health hospitals that began providing ART services earlier than 2017. Then, the proportional allocation was done for elected public hospitals based on the number of the study population. Furthermore, using medical record number at each hospital from June 27, 2017, to May 31, 2022, were considered, and a random number was generated using a computer. Finally, each study participant was selected using a computer-generated simple random sampling technique (Fig 1).

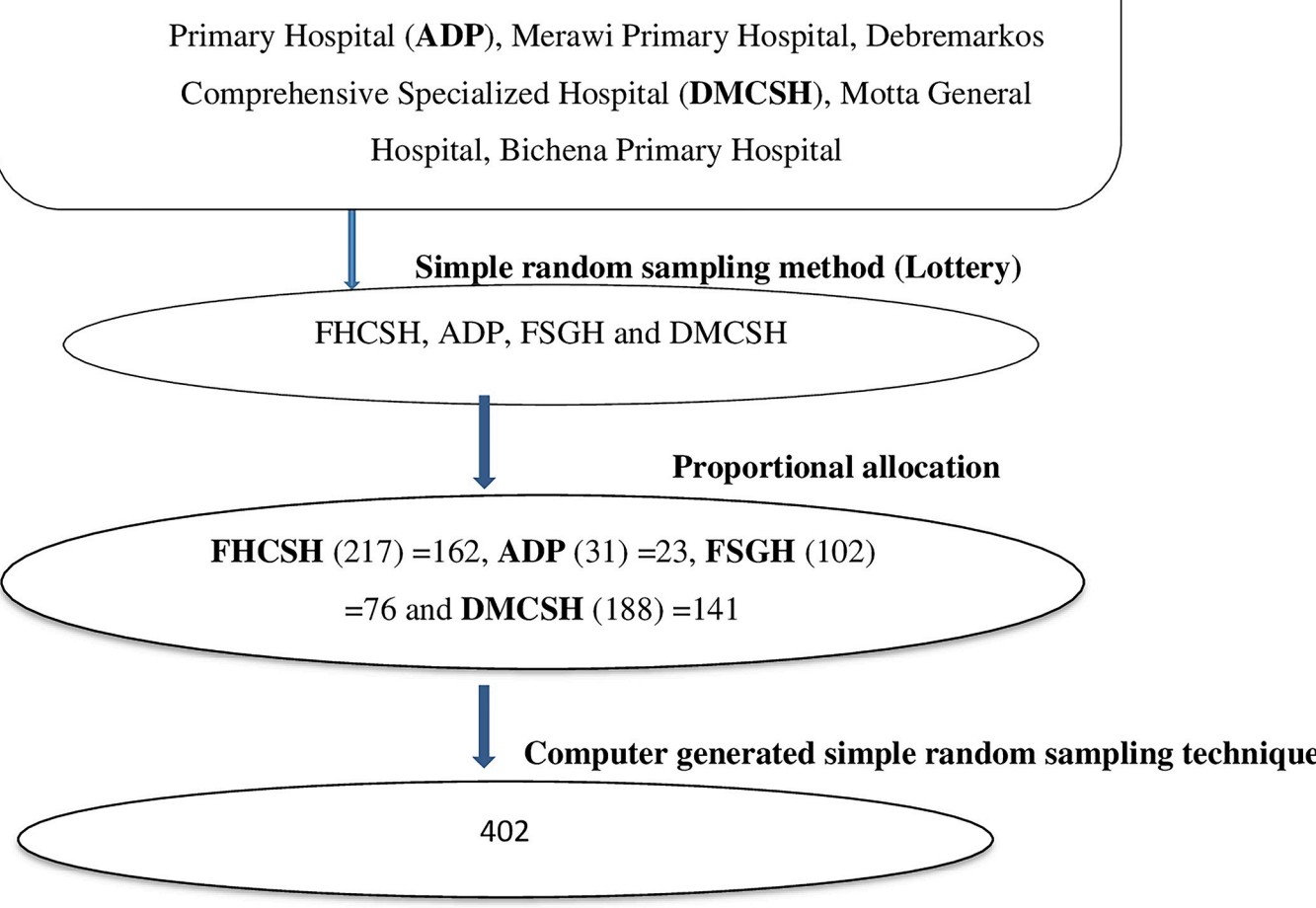

**Fig 1. Schematic diagram representation the sampling procedure of the study, northwest Amhara region public health hospital among children on ART, 2023.**

## Variables and operational definitions

**Dependent variable.** Time to develop major adverse drug reactions (MADRs).

## Independent variables

- **Socio-demographic Factors:** Age, sex, residence, parental aliveness, primary care giver, primary care giver HIV status, parent educational status

- **Clinical and Immunological Drug Factors:** Immune status, WHO clinical stage of HIV, chronic illness, opportunistic infections, base line CD4 count, base line HGB, base line functional status, baseline developmental status, BMI for age, baseline viral load, base line and weight for height.

- **ART Drug Regimen Type and Prophylaxis Related Factors:** Type of initial regimen, INH, Cotrimoxazole prophylaxis, treatment adherence, frequency dose.

**MADRs.** It is defined in this study as having any one of the features recorded as drug complaints about seeking care and resulted in either hospitalization, regimen change or switch/discontinued /and disability in the patient's body function: anemia, skin rash, elevated organ functional test, hepatotoxicity, CNS toxicity, and other rare conditions.

**Event.** A patient outcome who reported at least one of the listed MADRs.

**Censored.** Patients lost to follow up, dead and not develop MADRs till end of study.

**Functional status.** Was described as working, ambulatory, bedridden.

**Working.** Go to school, do normal activities, or play [24].

**Ambulatory.** Able to perform activities of daily living [24].

**Bedridden.** Not able to perform activities of daily living [24].

**Time to adverse drug reaction.** The time gap in months between being put on ART to the development of the first episode of ART adverse drug reactions [16].

**Good adherence.** If took $\geq$ 95% or ($<$2 doses of 30 doses or $<$3 doses of 60 doses is missed) asdocumented by the ART healthcare provider [16].

**Poor adherence.** If it is less than 85%, the patient is considered to have poor adherence if they missed more than six of every 30 doses or more than nine of every 60 doses [16].

**Appropriate development.** Milestones are things most children can do by a certain age. Skills such as taking a first step, smiling for the first time, and waving "bye-bye" are called developmental milestones [25].

**Delay development.** Is considered when a child takes longer to reach certain development milestones than other children their age [25].

**Developmental regression.** Is when a child who has reached a certain developmental stage begins to lose previously acquired milestones [26].

**Severe anemia.** (Hemoglobin $<$7g/dl in children aged 6–59 months and $<$8g/dl in children aged 5–14 years) [13].

## Data collection checklist and procedure

Charts were gathered at each hospital's ART clinic using their medical record numbers. Medical cards were collected and provided to data collectors with the assistance of card room staff. The data was extracted from each patient's record chart using a data extraction checklist prepared using reviewed literature and the ART treatment guideline [4, 23, 27]. Six (6) BSC nurses and four (4) MSc nurses worked as data collectors and supervisors, respectively.

### Data quality control

Nurses who took ART training served as data collectors. Both data collectors and supervisors were trained regarding data collection and the study's purpose. The supervisors and the primary investigator provided close supervision. A pretest was done using a 5% sample size (21) at DMCSH and FHCSH.

### Data analysis

Data was entered into Epi Data 4.6 and analyzed using SATA 14. Data was processed to generate the survival time to develop adverse drug reaction. Incidence rate MADRs was calculated per 1000 patients–months. Model fitness was checked using the log–log-rank graphical technique and Schoenfeld residual. A multivariable Cox regression hazard model was fitted for variable**s** with a p-value of $\leq 0.25$ in bivariable Cox proportional hazards regression model to identify predictors of the outcome variable. The adjusted hazard ratio (AHR) with a 95% confidence interval (CI) and a P-value of 0.05 was used to determine the strength of the association and statistical significance.

### Ethical approval

Ethical approval was secured from the research ethical committee of Debre Markos, College of Medicine and Health Sciences, with approval number MHSC/R/C/TT/D122/11/15. And also, a permission letter was secured from each hospital. Furthermore, because this was a retrospective study based on medical records of patents, individual assent was not applicable. The ethics committee formally waived for extraction of data from patient's medical records to meet the study's goal. Data collection started once hospital managers granted permission. All patient information was kept confidential.

## Results

### Sociodemographic characteristics

A total of 380 study participants' charts were included in the analysis. More than half (57.9% of the children were females. Nearly two third (64.2%) of children were in the age category of 5–15 years. The baseline age of the study participants was 11–180 months, and the mean age was 89.27 (SD + 2.42) months (**Table 1**).

### Baseline clinical, immunological, ART drug and prophylaxis related characteristics

More than one-third (36.5%) of study participants were in WHO clinical stage 2 of HIV/AIDS. More than half of the study participants (60.16%) used DTG-based ART regimens. Three hundred fifty-three (92.89%) participants had a CD4 count of more than or equal to 200 cells/mm3, and one hundred eleven (29.21%) of the study participants had OI. Almost the majority of the subjects (99.47%) had a viral load of 1000cop/mm3. Over 90% (354) of research participants received OI prophylaxis (**Table 2**).

### Incidence of major adverse drug reaction

The minimum and maximum follow-up times were 6 and 59 months, respectively, with a total follow-up time was 9916 months. Of the participants in the study, 35 (9.21%) experienced MADRs. The incidence rate of MADRs was 3.53 per 1000 person-months. Of those who experienced MADRs, 57.14%, 20%, and 17.14% of study participants experienced them in the 6th,

**Table 1. Baseline socio-demographic characteristics participants at selected public health selected public hospital, northwest Amhara, Ethiopia, 2023.** (n = 380).

| Variables | | Category | Frequency (%) |
|---|---|---|---|
| **Age** | | <24 months | 60 (15.79%) |
| | | 24–60 months | 76 (20.00%) |
| | | 60–120 months | 124 (32.63%) |
| | | ≥120 months | 120 (31.58%) |
| **Sex** | | Male | 160 (42.11%) |
| | | Female | 220 (57.89%) |
| **Residence** | | Urban | 168 (44.21%) |
| | | Rural | 212 (55.79%) |
| **Parental aliveness** | | Both alive | 290 (76.32%) |
| | | Mother died | 15 (3.95%) |
| | | Father died | 15 (3.95%) |
| | | Both died | 8 (2.11%) |
| | | Not known | 52 (13.68%) |
| **Primary care giver** | | Parent | 317 (83.42%) |
| | | None parent | 63 (16.58%) |
| **Marital status of caregiver** | | Married | 373 (98.16%) |
| | | Single | 7 (1.84%) |
| **HIV status of caregiver** | | Positive | 312 (82.11%) |
| | | Negative | 43 (11.32%) |
| | | Unknown | 25 (6.58%) |
| **Age of caregiver (years)** | | 20–35 | 223 (58.68%) |
| | | ≥35 | 157 (41.32%) |

12th, and 18th months. Anemia, severe skin rash, hepatotoxicity, CNS toxicity, renal problem, and others accounted for MADRs of 31.43%, 28.57%, 17.14%, 8.57%, 8.57%, and 5.71%, respectively.

The incidence rate of MADRs among females and males was 4.2/4278 and 3/5638 person months observation, respectively. The incidence rates of MADRs on ART medication regimens based on NVP, EFV, and DTG were 0.010/2232, 0.0028/3167, 0.0029/341, and 0.00071/4225 person-months observation time, respectively. Patients with good adherence had an incidence rate of 0.0022/9021, and those with fair or poor adherence had a MADRs incidence rate of 0.017/895 person months, respectively. The incidence rates of MADRs were 3/9055 and 8.1/860 person-months of observation, respectively, among individuals who received prophylaxis and those who did not. Incidence rates for MADRs were 0.0019/6192 and 0.0061/3724 person months, respectively, among persons in WHO clinical stages I, II, III, and IV (**S1 Table**).

## Median survival time among children on ART

The study's findings showed that children on ART had a median survival duration of 57 months. However, because the study had fewer events, restricted mean survival time was used, over the median survival time, which indicated the maximum event time [28]. The mean survival time using the restricted mean was 51.78 (95% CI: 49.89–53.66) months. However, among patients who had experienced the event (MADRs), the median survival time was 42 months (**Fig 2**).

**Table 2. Baseline clinical, immunological, ART drug and prophylaxis related characteristics participants at selected public health selected public hospital, northwest Amhara, Ethiopia, 2023.** (n = 380).

| Variables | Categories | Frequency (%) |
|---|---|---|
| **ART drug regimen** | DTG / Other based | 229 (60.26%) |
| | LPV/r based | 14 (3.68%) |
| | EFV based | 81 (21.32%) |
| | NVP based | 56 (14.74%) |
| **Baseline WHO stage** | Stage I | 107 (28.16%) |
| | Stage II | 137 (36.05%) |
| | Stage III | 116 (30.53%) |
| | Stage IV | 20 (5.26%) |
| **Opportunistic infection** | Yes | 111 (29.21%) |
| | No | 269 (70.79%) |
| **Baseline viral load** | <1000 | 378 (99.47%) |
| | ≥1000 | 2 (.53%) |
| **Baseline HGB level** | <11 | 7 (1.84%) |
| | 11–12 | 130 (34.21%) |
| | ≥12 | 243 (63.95%) |
| **Baseline CD4 count** | <200 | 27 (7.11%) |
| | ≥200 | 353 (92.89%) |
| **Developmental status age <5 years (110)** | Appropriate | 102 (92.73%) |
| | Delayed | 8 (7.27%) |
| **Functional status age (5–15 years)** | Working | 108 (40%) |
| | Ambulatory | 142 (52.59%) |
| | Bedridden | 20 (7.41%) |
| **Baseline weight for height** | > - 1 z-score | 79 (73.15%) |
| | < -1 and > -2 z- score | 22 (20.37%) |
| | < -2 and > -3 z-score | 2 (1.85%) |
| | < -3 z- score | 5 (4.63%) |
| **BMI for age (5–15 years** | > -1 z-score | 208 (76.47%) |
| | < -1 and > -2 z- score | 60 (22.06%) |
| | < -2 and > - 3 z- score | 4 (1.4%) |
| **OI prophylaxis given** | Yes | 354 (93.16%) |
| | No | 26 (6.84%) |
| **INH prophylaxis** | Yes | 260 (68.42%) |
| | No | 120 (31.58%) |
| **Adherence to ART drug** | Good adherence | 349 (91.84%) |
| | Fair / poor adherence | 31 (8.16%) |
| **ART drug intake frequency** | Once per day | 147 (38.68%) |
| | Twice or more per day | 233 (61.32%) |
| **Current status of patient** | Alive | 340 (89.47%) |
| | Dead and lost follow up | 10 (2.63%) |
| | Transfer out and transfer to adult | 30 (7.89%) |

## Assessing the proportional hazard assumption

According to the proportional hazard assumption, the study subjects' risk of failing must be constant over the course of their follow-up. The Schoenfeld residual assumption test (phtest) and a log-log probability plot of the graph were used to visually assess the model's fitness (chi-

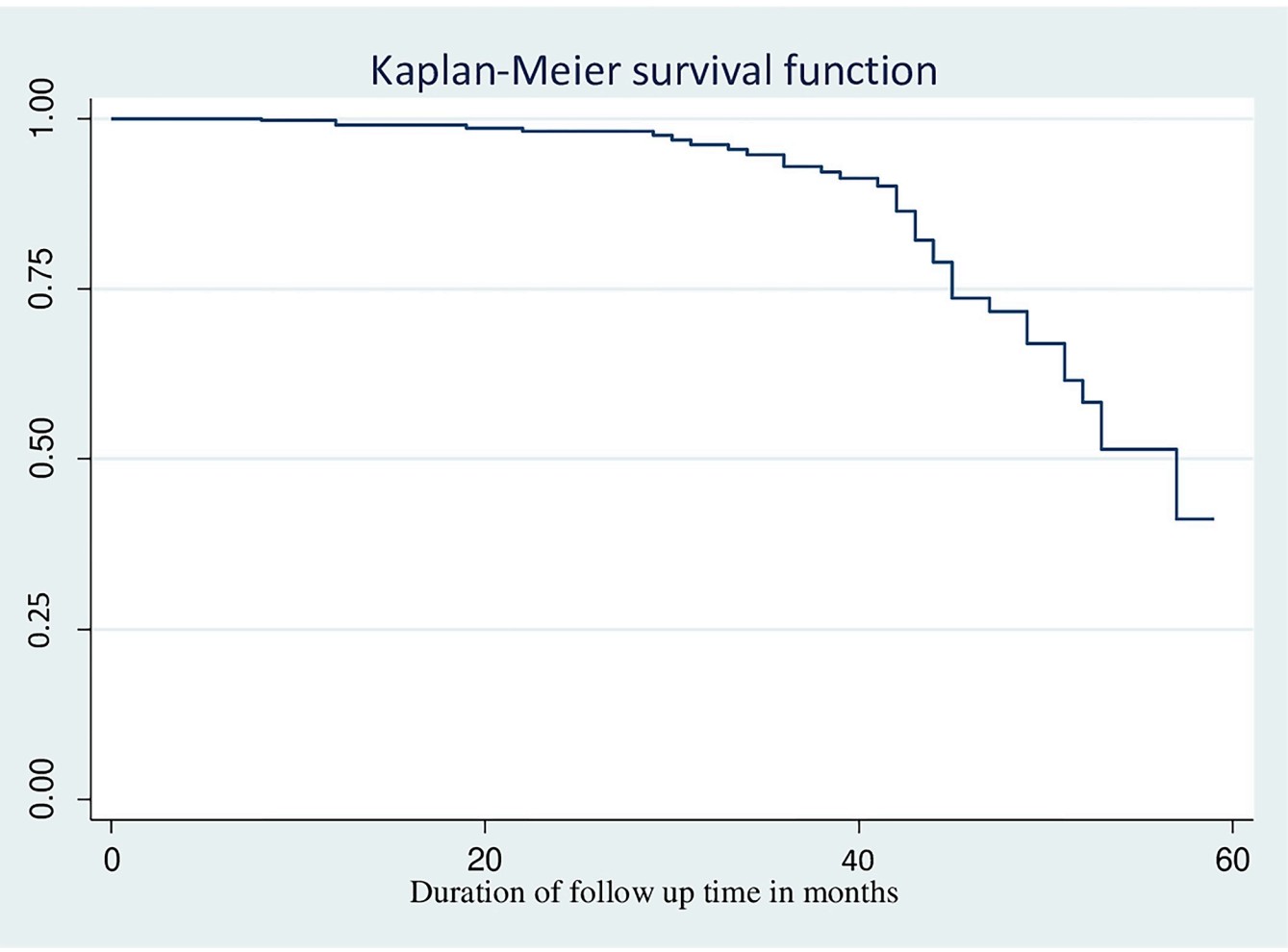

**Fig 2. Kaplan Meier curve estimate of overall survival status of children on ART drug at selected public hospital, northwest Amhara, Ethiopia, 2023.** (n = 380).

square = 4.81, P-value = 0.6826). The log-log probability plot of the graph and the phtest results showed that all required proportional hazard assumptions were satisfied (**S2 Table**).

## Comparative analysis of models

Semi-parametric and parametric proportional hazard models analysis was also done to select the most fitted model for overall model fitness to estimate the survival incidence of MADRs and its predictors among participants. Using information criterion (AIC) and log-likelihood results. Based on all three comparisons, the Gompertz regression model chosen to have a lower AIC value (AIC = 66.78037, BIC = 114.0624, log likelihood = -21.390184) was more efficient than Cox-PH and other parametric models (**S3 Table**).

## Kaplan Meier survival curves

Kaplan-Meier survival curve showed the survival difference between groups and the timing of the MADRs in the survival rates. The difference in the research participants' baseline WHO clinical stage of HIV/AIDS III& IV was associated with a significant increase in the development of MADRs compared to counterparts (long rank test chi-square = 9.91 and p-

**Table 3. Bivariate and multivariate cox regression analysis predictor variables of time to the development of MADRs participants at selected public health selected public hospital, northwest Amhara, Ethiopia, 2023.** (n = 380).

| Variables | Categories | Survival status | | CHR(95%CI) | AHR(95%CI) | P-value |
|---|---|---|---|---|---|---|
| | | Event | Censored | | | |
| Adherence to ART drug | Good adherence | 21 | 328 | 1.00 | 1.00 | |
| | Fair/poor adherence | 14 | 17 | 9.74(4.8–19.39) | 6.40(3.14–13.04) | 0.001 |
| Baseline WHO clinical stage of HIV | I and II | 11 | 233 | 1.00 | 1.00 | |
| | III and IV | 24 | 112 | 2.90(1.44–5.86) | 3.51(1.41–8.76) | 0.007 |
| Taking OI Prophylaxis | Yes | 29 | 325 | 1.00 | 1.00 | |
| | No | 6 | 20 | 2.64(1.14–6.09) | 3.09(1.17–8.16) | 0.022 |
| ART drug intake frequency | Once per day | 6 | 141 | 1.00 | 1.00 | |
| | ≥2 times per day | 29 | 204 | 2.52(0.97–6.54) | 3.43(1.26–9.33) | 0.016 |
| Being infected with OI | Yes | 14 | 97 | 1.00 | 1.00 | |
| | No | 21 | 248 | 0.81(0.33–1.30) | 0.65(0.39–1.86) | 0.697 |

value = 0.0016) (S1 Fig). This study also reported that children with good adherence status had a better survival rate than fair/poor adherence to ART (S2 Fig). In addition, this study found that a difference in time to MADRs was observed in children who took ART drugs once per day and those who took twice or more (S3 Fig). Finally, this study found that there is a significant difference in time to MADRs between children on ART who received OI prophylaxis and those who did not (S4 Fig).

## Predictors of time to MADRs among HIV positive children on ART

OI prophylaxis, INH prophylaxis, adherence to ART drug, ART drug intake frequency, WHO clinical stage of disease, and OI infection were associated with time to MADRs in the bivariate cox-regression model with a p-value less than or equal to 0.25. In multivariable cox-regression, only predictors such as adherence to ART drug, ART drug intake frequency, WHO clinical stage of disease, and OI prophylaxis remained statistically significant at a p-value less than 0.05.

This study revealed that the hazard-developing MADRs among patients with advanced WHO clinical stage of HIV/AIDS was 3.51 times higher than stage I and II patients [AHR = 3.51, (95% CI, 1.41–8.76)]. The hazard of developing MADRs among patients who took ART drugs twice or more per day was 3.43 times more likely at risk of developing MADRs than those who once per day [(AHR = 3.43, (95%CI, (1.26–9.33)]. And also, this study revealed that the hazard of developing MADRs among patients with fair or poor adherence is 6.40 times higher risk compared to counterparts [(AHR = 6.40, (95%CI, (3.14–13.04)]. Finally, this found that patients who didn't take prophylaxis for opportunistic were 3.09 times more likely to develop MADRs compared to their counterparts [AHR = 3.09, (95%CI, 1.17–8.16)] (Table 3).

## Discussion

The effect of adverse drug reactions on antiretroviral treatment efficacy depends on the type and severity of the adverse drug reactions (ADRs) and the timing of the adverse drug reaction. The timing of adverse drug reactions in HIV patients is a critical public health concern for ART treatment adherence and retention. Therefore, this study aimed to assess the time to MADRs and its predictors among HIV/AIDS-positive children on ART. The study reported that 9.21% of participants developed MADRs with an incidence rate of 3.5/1000 (PM) person

per month of follow-up with 95% CI [2.5–4.9]. This finding is consistent with studies in Ethiopia and Nigeria [15, 29–31] supported this evidence.

In this study, females had a higher incidence of MADRs than males, with 4.2/4278 and 3/5638, respectively. This finding is consistent with studies in Debre Markos (Ethiopia), and Mali [30, 32]. This might be due to sex-related differences in adverse drug reactions (ADRs) to antiretroviral drugs, where females report ADRs at a higher rate than males beginning at puberty [33, 34]. The incidence of MADRs among NVP-based ART drug regimens was higher than EFV-based ART, which was 0.010/2232 and 0.0028/3167, respectively. Studies in Ethiopia supported this evidence [27, 30]. This is due to the lower rate of severe adverse drug reactions in EFV compared to NFV, particularly treatment discontinuations [35].

This study reported that the cumulative probability of surviving without developing MADRs was 0.99, 0.99, 0.95, 0.83, and 0.85 in 1st year, 2nd year, 3rd year, 4th year, and 5th year, respectively, and the cumulative survival without experiencing MADRs is 92.2%. This result is higher than studies from Ethiopia [16, 29, 30]. This could be due to the current recommendation of DTG-based ART drug regimens rather than previous NVP and EFV-based regimens because DTG-based drug regimens are less toxic than NVP and EFV-based regimens. Newer ART drug regimens, such as DTG-based regimens, are associated with fewer adverse drug effects [36].

This study revealed that patients in the advanced clinical stage at the initiation of ART had a 3.85 times higher risk of developing MADRs at any time compared to patients in clinical stages II and I. The finding is consistent with studies in Ethiopia, where clients in the advanced clinical stage of the disease were more likely to develop MADRs [23, 30, 37, 38]. Patients at advanced clinical stages of HIV/AIDS often had significant immunosuppression, characterized by lower CD4 counts. This immunocompromised state can increase susceptibility to infections and raise the occurrence of adverse reactions due to the body's decreased ability to handle the stress of new ART drugs. Many children reporting adverse drug reactions had CD4 counts $< 300$ cells/mm$^3$, indicating a link between immunosuppression and the occurrence of adverse drug reactions [39]. In addition, advanced clinical stages are often associated with high viral loads. A high viral load can have an impact on the pharmacodynamics of ART, perhaps increasing toxicity or causing adverse effects as the body fights to handle both the viral infection and the side effects of the drugs. This dual burden can exacerbate the risk of experiencing MADRs.

Furthermore, children with advanced stages may require more complex ART regimens, including combinations of multiple drugs, to adequately treat their illness. The complexity of these regimens raises the risk of drug-drug interactions and adverse reactions, as the possibility of suffering side effects is often associated with the number of medications taken concurrently.

The hazard of developing MADRs among children who did not take OI prophylaxis was 3.09 times higher risk compared to their counterparts. Studies in Bahirdar and North West Amhara Specialized Hospitals [23, 37] supported this finding. Children without OI prophylaxis are at a greater risk of developing opportunistic infections. The presence of OI infections can complicate treatment and increase the likelihood of MADRs, as the body may react more severely to the combination of ART and the stress of an ongoing infection.

Children with poor adherence to ART drugs had a 6.4 higher risk of experiencing MADRs than patients with good adherence. A study in West Hararghe Zone supported this finding [16]. This could be due to children with poor adherence to ART drugs are more likely to develop co-morbidities or OI as a result of weakened immunity, which puts them at a higher risk of MADRs due to drug-drug interactions when taking extra drugs to treat OI infection. Poor adherence results in inconsistent and low therapeutic drug levels, which can lead to resistance and raise the risk of MADRs due to the use of alternative drugs with distinct adverse

effect profiles, potentially increasing the risk of ADRs. Children with poor adherence may face various psychological challenges, including a lack of support, stigma, or a misunderstanding of their treatment plan. These conditions can exacerbate the likelihood of experiencing ADRs since they may not disclose side effects or seek care soon, resulting in more adverse severe reactions.

Finally, this study revealed that the risk of developing MADRs in children who took ART >2(two) times/ day increased by 43% compared to those who took once/ day. This study is supported by national ART treatment guideline [22]. This could be because patients who take ART medications twice a day or more are more prone to forget their drug intake dose, which results in decreased immunity and MADRs.

The finding has important clinical implications for reducing the incidence of MADRs by enhancing patients' ART adherence to prevent opportunistic infection and advanced-stage HIV/AIDS. The finding is also important for policymakers in developing strategies and integrated clinical intervention to prevent poor treatment adherence, the occurrence of opportunistic infection, and the advanced stage of HIV/AIDS. Finally, it had public health importance by preventing economic loss associated with HIV/AIDS and adverse drug reactions.

## Conclusions

The overall incidence rate MADRs among children on ART was "3.5/1000" person—months. Advanced WHO clinical stage, poor adherence to ART drugs, drug intake frequency, and taking OI prophylaxis were associated with the incidence of MADRs. Therefore, especial emphasis should be given, to patients with advanced clinical stages of HIV/AIDS, poor adherence to ART, and not taking prophylaxis. And also responsible should implement strategies to improve the quality of care given to such patients to prevent poor ART adherence, opportunistic infection, and advanced stage of HIV/AIDS.

## Limitation of the study

Because the study was based secondary data, important variables such sociodemographic and clinical characteristics, may be missed due to poor documentation. Additionally, records and measurements made by many people increase the potential of observer bias.

## Supporting information

**S1 Checklist.**
(DOCX)

**S1 Fig. Kaplan Meier curve showing time to the development of MADRs of participants based on WHO clinical stage of HIV/AIDS at selected public hospitals, northwest Amhara, Ethiopia, 2023.** (n = 380).
(TIF)

**S2 Fig. Kaplan Meier curve showing time to the development of MADRs of participants based on ART adherence at selected public hospitals, northwest Amhara, Ethiopia, 2023.** (n = 380).
(TIF)

**S3 Fig. Kaplan Meier curve showing time to the development of MADRs of participants based on ART frequency intake at selected public hospitals, northwest Amhara, Ethiopia, 2023.** (n = 380).
(TIF)

**S4 Fig. Kaplan Meier curve showing time to the development of MADRs of participants based on intake of OI prophylaxis at selected public hospitals, northwest Amhara, Ethiopia, 2023.** (n = 380).
(TIF)

**S1 Table. Life table for MADRs survival among children on ART at selected public health selected public hospital, northwest Amhara, Ethiopia, 2023.** (n = 380).
(DOCX)

**S2 Table. Global test result of variables among HIV positive children on ART, at selected public hospital northwest Amhara, Ethiopia, 2023.** (n = 380).
(DOCX)

**S3 Table. Parametric and semi parametric model comparison among HIV positive children on ART, at selected public hospital northwest Amhara, Ethiopia, 2023.** (n = 380).
(DOCX)

## Acknowledgments

We are grateful to Debre Markos University, the College of Medicine and Health Sciences, and the individuals who participated in the study for their cooperation and technical support.

## Author Contributions

**Conceptualization:** Bantegizie Senay Tsega, Abebe Habtamu, Moges Wubie.

**Data curation:** Bantegizie Senay Tsega, Keralem Anteneh Bishaw.

**Formal analysis:** Bantegizie Senay Tsega, Worku Misganaw Kebede, Keralem Anteneh Bishaw.

**Investigation:** Animut Takele Telayneh, Bekalu Endalew.

**Methodology:** Bantegizie Senay Tsega, Samuel Derbie Habtegiorgis, Molla Yigzaw Birhanu, Keralem Anteneh Bishaw.

**Supervision:** Abebe Habtamu, Moges Wubie, Animut Takele Telayneh.

**Validation:** Bekalu Endalew, Samuel Derbie Habtegiorgis, Molla Yigzaw Birhanu.

**Writing – original draft:** Bantegizie Senay Tsega, Keralem Anteneh Bishaw.

**Writing – review & editing:** Worku Misganaw Kebede, Keralem Anteneh Bishaw.

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
