## [Decision Letter · Decision Letter 0]

17 Jul 2024

PONE-D-23-20912TIME TO MAJOR ADVERSE DRUG REACTIONS AND ITS PREDICTORS AMONG CHILDREN ON ANTIRETROVIRAL TREATMENT AT NORTHWEST AMHARA  SELECTED PUBLIC HOSPITALS  NORTHWEST; ETHIOPIA, 2023.PLOS ONE

Dear Dr. Bishaw,

Thank you for submitting your manuscript to PLOS ONE. After careful consideration, we feel that it has merit but does not fully meet PLOS ONE’s publication criteria as it currently stands. Therefore, we invite you to submit a revised version of the manuscript that addresses the points raised during the review process.

We look forward to receiving your revised manuscript.

Kind regards,

Edmund Tetteh Nartey

Academic Editor

PLOS ONE

5. Please include your tables as part of your main manuscript and remove the individual files. Please note that supplementary tables (should remain/ be uploaded) as separate "supporting information" files

Reviewers' comments:

Reviewer's Responses to Questions

**Comments to the Author**

1. Is the manuscript technically sound, and do the data support the conclusions?

Reviewer #1: Yes

Reviewer #2: Yes

2. Has the statistical analysis been performed appropriately and rigorously? 

Reviewer #1: Yes

Reviewer #2: I Don't Know

3. Have the authors made all data underlying the findings in their manuscript fully available?

Reviewer #1: Yes

Reviewer #2: Yes

4. Is the manuscript presented in an intelligible fashion and written in standard English?

Reviewer #1: Yes

Reviewer #2: Yes

5. Review Comments to the Author

Reviewer #1: My Review Comments

The researchers retrospectively studied time to major adverse drug reactions and its predictors among children on antiretroviral treatment in selected public hospitals in Ethiopia. This study is quite interesting because adverse drug reactions (ADRs) are common in HIV treatment and as the authors rightly put it, determining the incidence rate and predictors of ADRs among children on ART may be useful in improving treatment outcomes and minimizing harm. The authors indicated limited evidence regarding the time to ADRs and its predictors among children on ART in Ethiopia, and this further justified the study.

The researchers employed a structured data extraction checklist to collect data for the study. They further indicated how data was analyzed and how parameters of interest were determined. The authors determined the incident rate of major ADRs per person/month, while they used the Cox proportional hazards regression model to identify predictors of major adverse drug responses. They declare statistical significance when p-values of less than 0.05 at 95% CI were attained.

The researchers followed the participants up for between 6 and 59 months for a total of 9916 person-months and observed an incidence rate of major ADRs of 3.5 / 1000 person-months. From the results of the study, the researchers reported congruence of their observed incidence rate of major ADRs among children on ART with previous studies in Ethiopia. They further reported advanced clinical stages of HIV / AIDS, poor treatment adherence, taking ART medications twice or more, and not taking opportunistic infection prophylaxis as predictors of major ADRs.

Introduction

The introduction was quite well-written. The authors gave some statistics on the global and Sub-Saharan Africa prevalence of HIV/AIDS with some useful information on ensuing ADRs during treatment of affected patients. The authors then narrowed down on similar studies carried out in Namibia, Nigeria, and in Ethiopia.

Method

The study area, setting, and population were well defined, as well as the duration of the recruitment of study participants retrospectively. The data extraction period was also specified to be from February 15 to March 30/2023.

The researchers stated how the sample test for the study was determined. The authors also explained the sampling technique and sampling procedure used to enroll the study participants and this was quite appropriate. Variables and operational definitions were provided by the authors. The authors stated a data collection checklist and procedure that is quite sound. They also included how data quality control was achieved. The researchers gave details on how data was analyzed.

The authors secured ethical clearance for the study and included the approval number. They also secured permissions from the study sites.

Results

A table and description of social demographic characteristics were provided. Under baseline clinical, immunological, ART drug and prophylaxis related characteristics, the researchers stated that more than half of the study participants (60.16%) used DTG-based ART regimens but failed to show what ART drug regimen the remaining 39.84% of the study participants used.

Minor Revision

Abstract

1. Authors should write ART, and AHR in full at first use.

Introduction

2. Paragraph 3 line 5: Authors should write AVR in full at first use.

3. Paragraph 5 line 4: Authors should write AZT in full at first use.

4. Paragraph 6 line 4: Authors should write BMI, and TDF-3TC-EFV in full at first use.

5. Paragraph 7 line 3: Authors should write TDF-NVP in full at first use.

6. Paragraph 8 line 2: Authors should write MADRs, and DTG in full at first use.

Methods

7. Authors should write all abbreviations in full at first use in all sections of the manuscript.

Results

8. Under baseline clinical, immunological, ART drug and prophylaxis related characteristics, the researchers stated that "more than half of the study participants (60.16%) used DTG-based ART regimens" but failed to show what ART drug regimen the remaining 39.84% of the study participants used. The authors should include this data and discuss any observed effect the different ART regimen may have on time to major ADRs in study participants.

Discussion

The discussion was well written.

9. However, the authors compared their findings with previous findings but gave no brief description of such findings. The authors should include very brief description of the previous findings to make their comparison more meaningful.

Reviewer #2: Opening Comments:

This is an important by Bantegizie Senay and colleagues. It sought to retrospectively examine adherence of children infected with HIV to ARTs’ in Amhara in Ethiopia.

The manuscript is well written...

Title:

Time to Major Adverse Drug Reactions and Its Predictors Among Children on Antiretroviral Treatment at Northwest Amhara Selected Public Hospitals Northwest; Ethiopia, 2023.

Abstract

- The abstract is ok, and well structured.

Introduction

- Paragraph 5, line 5: Revise ‘4.18–2.3 gm/dl’ to read 4.2 -2.3 g/dl

- Paragraph 6, line 2: Revise ….’majority of (90.74%)’….to read ….’majority (90.7%) of’ …., and delete the last 2 words (i.e. of ART ) at the end of that sentence.

- Last Paragraph, second line: Write the acronym ‘MADRs’ in full, and put the MADRs in parentheses or bracket here

Methods

Study area, setting, and population

- Revise the first sentence into 2 sentences

- Line 6 and 7, add the respective hospitals to complete that sentence

Data Quality Control

- Last sentence, from Line 4: Should read 5% with the number obtained in parentheses

- Write DMCSH and FHCSH in full

Data Analysis

- Line 2: Correct the error in the acronym MADR’s from MDAR’s

Results

Incidence of major adverse drug reaction

- Line 1, Paragraph 2: Kindly take a second look at the sentence ‘The incidence rate of MADR among females and males was 4.2/4278 and 3/5638 person months’ in comparison with the female: male population in Table 1, Review and confirm that its collaborative and correct

Discussion

- Line 5: Should read…. The study reports that ….

- Second Paragraph, Line 2: Revise to read: ‘in Debre Markos (in Ethiopia) and Mali’.

Conclusions

- Revise the first sentence, especially the quotation of the value ‘3.5/1000’.

Summary

1. Manuscript requires minor revisions indicated.

Thanks

6. PLOS authors have the option to publish the peer review history of their article (what does this mean?). If published, this will include your full peer review and any attached files.

Reviewer #1: No

Reviewer #2: No

---

## [Author Response · Author response to Decision Letter 0]

15 Aug 2024

Date: August 13, 2024

Manuscript ID: PONE-D-23-20912

Title: Time to Major Adverse Drug Reactions and Its Predictors among Children on Antiretroviral Treatment at Northwest Amhara Selected Public Hospitals Northwest; Ethiopia, 2023.

Response to editors 

Dear Editors: 

Thank you for sending us your valuable comments, which immensely improved our manuscript. We included all the editorial comments raised and we also enclosed the point by point response of six (6) pages attached here with.

It is my pleasure to inform you that the manuscript was edited meticulously by the English Language expert from Debre Markos University - Ethiopia. 

Respectfully, 

Bishaw, KA

Corresponding author 

Reviewer #1: My Review Comments

The researchers retrospectively studied time to major adverse drug reactions and its predictors among children on antiretroviral treatment in selected public hospitals in Ethiopia. This study is quite interesting because adverse drug reactions (ADRs) are common in HIV treatment and as the authors rightly put it, determining the incidence rate and predictors of ADRs among children on ART may be useful in improving treatment outcomes and minimizing harm. The authors indicated limited evidence regarding the time to ADRs and its predictors among children on ART in Ethiopia, and this further justified the study.

The researchers employed a structured data extraction checklist to collect data for the study. They further indicated how data was analyzed and how parameters of interest were determined. The authors determined the incident rate of major ADRs per person/month, while they used the Cox proportional hazards regression model to identify predictors of major adverse drug responses. They declare statistical significance when p-values of less than 0.05 at 95% CI were attained.

The researchers followed the participants up for between 6 and 59 months for a total of 9916 person-months and observed an incidence rate of major ADRs of 3.5 / 1000 person-months. From the results of the study, the researchers reported congruence of their observed incidence rate of major ADRs among children on ART with previous studies in Ethiopia. They further reported advanced clinical stages of HIV / AIDS, poor treatment adherence, taking ART medications twice or more, and not taking opportunistic infection prophylaxis as predictors of major ADRs.

Introduction

The introduction was quite well-written. The authors gave some statistics on the global and Sub-Saharan Africa prevalence of HIV/AIDS with some useful information on ensuing ADRs during treatment of affected patients. The authors then narrowed down on similar studies carried out in Namibia, Nigeria, and in Ethiopia.

Method

The study area, setting, and population were well defined, as well as the duration of the recruitment of study participants retrospectively. The data extraction period was also specified to be from February 15 to March 30/2023.

The researchers stated how the sample test for the study was determined. The authors also explained the sampling technique and sampling procedure used to enroll the study participants and this was quite appropriate. Variables and operational definitions were provided by the authors. The authors stated a data collection checklist and procedure that is quite sound. They also included how data quality control was achieved. The researchers gave details on how data was analyzed.

The authors secured ethical clearance for the study and included the approval number. They also secured permissions from the study sites.

Results

A table and description of social demographic characteristics were provided. Under baseline clinical, immunological, ART drug and prophylaxis related characteristics, the researchers stated that more than half of the study participants (60.16%) used DTG-based ART regimens but failed to show what ART drug regimen the remaining 39.84% of the study participants used.

Minor Revision

Response: Dear reviewer, thank you the constructive comment.

Abstract

Comment 1: Authors should write ART, and AHR in full at first use.

Response: Dear reviewer, correction was made accordingly and highlighted in the main document

Introduction

Comment 2. Paragraph 3 line 5: Authors should write AVR in full at first use.

Response: Dear, correction was made accordingly and highlighted in the main document

Comment 3. Paragraph 5 line 4: Authors should write AZT in full at first use.

Response: Dear, correction was made accordingly and highlighted in the main document

Comment 4. Paragraph 6 line 4: Authors should write BMI, and TDF-3TC-EFV in full at first use.

Response: Dear, correction was made accordingly and highlighted in the main document

Comment 5. Paragraph 7 line 3: Authors should write TDF-NVP in full at first use.

Response: Dear, correction was made accordingly and highlighted in the main document

Comment 6. Paragraph 8 line 2: Authors should write MADRs, and DTG in full at first use.

Response: Dear, correction was made accordingly and highlighted in the main document

Methods

Comment 7. Authors should write all abbreviations in full at first use in all sections of the manuscript.

Response: Dear, correction was made accordingly and highlighted in the main document

Results

Comment 8. Under baseline clinical, immunological, ART drug and prophylaxis related characteristics, the researchers stated that "more than half of the study participants (60.16%) used DTG-based ART regimens" but failed to show what ART drug regimen the remaining 39.84% of the study participants used. The authors should include this data and discuss any observed effect the different ART regimen may have on time to major ADRs in study participants.

Response: Dear, thank for the constructive comment. Discussion based on this idea was considered and included in the main document.

Discussion

The discussion was well written.

Comment 9. However, the authors compared their findings with previous findings but gave no brief description of such findings. The authors should include very brief description of the previous findings to make their comparison more meaningful.

Response: Dear, thank for the constructive comment. Correction was considered and included in the main document to describe the finding of the study briefly. 

Reviewer #2: Opening Comments:

This is an important by Bantegizie Senay and colleagues. It sought to retrospectively examine adherence of children infected with HIV to ARTs’ in Amhara in Ethiopia.

The manuscript is well written...

Title: Time to Major Adverse Drug Reactions and Its Predictors among Children on Antiretroviral Treatment at Northwest Amhara Selected Public Hospitals Northwest; Ethiopia, 2023.

Abstract: The abstract is ok, and well structured.

Introduction

Comment1: Paragraph 5, line 5: Revise ‘4.18–2.3 gm/dl’ to read 4.2 -2.3 g/dl.

Response: Dear, thank for the constructive comment. Correction was considered and highlighted in the main document as 4.2 -2.3 g/dl.

Comment2: Paragraph 6, line 2: Revise ….’majority of (90.74%)’….to read ….’Majority (90.7%) of’ …., and delete the last 2 words (i.e. of ART) at the end of that sentence.

Response: Dear correction was considered and the term “of ART” was deleted and included in the main document as “A study in Ethiopia reported that more than ninety percent (90.74%) of participants developed ADRs within one (1) year” 

Comment3: Last Paragraph, second line: Write the acronym ‘MADRs’ in full, and put the MADRs in parentheses or bracket here.

Response: Dear, correction was made and included as major adverse drug reactions “‘MADRs” in the main document.

Methods

Study area, setting, and population

Comment 4: Revise the first sentence into 2 sentences

Response: Thank you for the comment. Correction was considered and included in the main document.

Comment 5: - Line 6 and 7, add the respective hospitals to complete that sentence

Response: Correction was considered and respective hospitals (Felege Hiwot Comprehensive Specialized Hospital (FHCSH), Adet Primary Hospital, Finote Selam General Hospital, and Debre Markos Comprehensive Specialized Hospital (DMCSH)) were included in the main document.

Data Quality Control

Comment 6: - Last sentence, from Line 4: Should read 5% with the number obtained in parentheses

Response: Correction was considered and included in the main document.

Comment 7: Write DMCSH and FHCSH in full. 

Response: Correction was considered and included in the main document

Data Analysis

Comment 8: Line 2: Correct the error in the acronym MADR’s from MDAR’s 

Response: Correction was considered and included as “MADR’s” in the main document.

Results

Incidence of major adverse drug reaction

Comment 9: Line 1, Paragraph 2: Kindly take a second look at the sentence ‘the incidence rate of MADR among females and males was 4.2/4278 and 3/5638 person months’ in comparison with the female: male population in Table 1, Review and confirm that it’s collaborative and correct.

Response: Thank you dear for the comment. I checked this sentence “The incidence rate of MADRs among females and males was 4.2/4278 and 3/5638 person months’ in comparison with the female: male population in Table 1”. Theses sentence is correct and considered as it is in the main document. It reported the incidence rate MADRs in since it is the difference in the incidence rate between male and female individuals. 

Discussion

Comment 10: Line 5: Should read…. The study reports that ….

Response: Dear, correction was considered. 

Comment11: - Second Paragraph, Line 2: Revise to read: ‘in Debre Markos (in Ethiopia) and Mali’.

Response: Dear, correction was considered and included in the main document.

Conclusions

Comment 12:- Revise the first sentence, especially the quotation of the value ‘3.5/1000’.

Response: Correction was considered and included in the main document

Summary

1. Manuscript requires minor revisions indicated.

Thanks

Respectfully, 

Bishaw, KA

Corresponding author

---

## [Editor Report · Decision Letter 1]

20 Aug 2024

TIME TO MAJOR ADVERSE DRUG REACTIONS AND ITS PREDICTORS AMONG CHILDREN ON ANTIRETROVIRAL TREATMENT AT NORTHWEST AMHARA  SELECTED PUBLIC HOSPITALS  NORTHWEST; ETHIOPIA, 2023.

PONE-D-23-20912R1

Dear Dr. Bishaw,

We’re pleased to inform you that your manuscript has been judged scientifically suitable for publication and will be formally accepted for publication once it meets all outstanding technical requirements.

Kind regards,

Edmund Tetteh Nartey

Academic Editor

PLOS ONE
---

## [Editor Report · Acceptance letter]

24 Sep 2024

PONE-D-23-20912R1 

PLOS ONE

Dear Dr. Bishaw, 

I'm pleased to inform you that your manuscript has been deemed suitable for publication in PLOS ONE. Congratulations! Your manuscript is now being handed over to our production team.

Kind regards, 

on behalf of

Dr. Edmund Tetteh Nartey 

Academic Editor

PLOS ONE